

# Regional distribution of *Christensenellaceae* and its associations with metabolic syndrome based on a population-level analysis

Xiang Li[*], Zewen Li[*], Yan He, Pan Li, Hongwei Zhou and Nianyi Zeng

Microbiome Medicine Center, Division of Laboratory Medicine, Zhujiang Hospital, Southern Medical University, Guangzhou, Guangdong province, China

[*] These authors contributed equally to this work.

## ABSTRACT

The link between the gut microbiota and metabolic syndrome (MetS) has attracted widespread attention. *Christensenellaceae* was recently described as an important player in human health, while its distribution and relationship with MetS in Chinese population is still unknown. This study sought to observe the association between *Christensenellaceae* and metabolic indexes in a large sample of residents in South China. A total of 4,781 people from the GGMP project were included, and the fecal microbiota composition of these individuals was characterized by 16S rRNA sequencing and analyzed the relation between *Christensenellaceae* and metabolism using QIIME (Quantitative Insight Into Microbial Ecology, Version 1.9.1). The results demonstrated that microbial richness and diversity were increased in the group with a high abundance of *Christensenellaceae*, who showed a greater complexity of the co-occurrence network with other bacteria than residents who lacked *Christensenellaceae*. The enriched bacterial taxa were predominantly represented by *Oscillospira*, *Ruminococcaceae*, *RF39*, *Rikenellaceae* and *Akkermansia* as the *Christensenellaceae* abundance increased, while the abundances of *Veillonella*, *Fusobacterium* and *Klebsiella* were significantly reduced. Furthermore, *Christensenellaceae* was negatively correlated with the pathological features of MetS, such as obesity, hypertriglyceridemia and body mass index (BMI). We found reduced levels of lipid biosynthesis and energy metabolism pathways in people with a high abundance of *Christensenellaceae*, which may explain the negative relationship between body weight and *Christensenellaceae*. In conclusion, we found a negative correlation between *Christensenellaceae* and MetS in a large Chinese population and reported the geographical distribution of *Christensenellaceae* in the GGMP study. The association data from this population-level research support the investigation of strains within *Christensenellaceae* as potentially beneficial gut microbes.

## INTRODUCTION

Metabolic syndrome (MetS) is a disease that brings together a variety of metabolic disorders. With economic development and the improvement of living standards, the incidence of

Corresponding authors
Hongwei Zhou, hzhou@smu.edu.cn
Nianyi Zeng, zengny1@i.smu.edu.cn

MetS has also gradually increased, making this disease a global public health problem. MetS is characterized by abdominal obesity, dyslipidemia, hypertension, and elevated blood sugar, and it currently affects approximately 20 to 30% of adults worldwide (*Grundy, 2008*). There are several factors involved in the development of the disease, including host genetic factors (*Kraja et al., 2011*), eating habits and sedentary lifestyle (*He et al., 2018*), but the pathogenesis of MetS has not been fully elucidated.

With technological development, the role of the microbiome in human health has been increasingly emphasized. The gut microbiota impacts a range of human health conditions, including metabolic processes, immune-related diseases and neurological disorders (*Integrative HMP Research Network Consortium, 2019*; *Festi et al., 2014*; *Fung, Olson & Hsiao, 2017*). Studies on MetS and the gut microbiota in mouse models have shown that the development of MetS involves a combination of the gut microbiota, host genes and diet (*Ussar et al., 2015*; *Zmora, Suez & Elinav, 2019*). Specific gut microbiota, bacterial metabolic pathways and their interactions with human health are new focuses of microbiome research (*Proctor, 2019*), and understanding of the microbiome is intended to pave the way for future microbiological therapies (*Douillard & De Vos, 2019*). An increasing number of studies have reported that probiotics in the intestinal tract can maintain intestinal homeostasis by regulating glucose and lipid metabolism, inhibiting the inflammatory response, and improving metabolic disorders (*Lau et al., 2019*; *Plaza-Díaz et al., 2017*; *Sanders et al., 2019*), thus preventing MetS and related complications.

In this study, we focused on a family of *Firmicutes* named *Christensenellaceae,* where the type strain *Christensenellaceae minuta* was first isolated from the feces of a healthy Japanese man in 2012 and named after the Danish microbiologist Henrik Christensen (*Morotomi, Nagai & Watanabe, 2012*; *Waters & Ley, 2019*). Since this family of bacteria was recently isolated, little is known about its biological function other than its association with the host and with other microorganisms. *Goodrich et al. (2014)* found that *Christensenellaceae* accounted for 0.01% of human feces from the UK Twins population. In data from China (*Kong et al., 2016*; *Wang et al., 2015*), South Korea (*Kim et al., 2019*) and Italy (*Biagi et al., 2016*), the bacteria were found to be highly abundant in centenarians. Brooks et al. reported that American females have a higher abundance of *Christensenellaceae* than American males (*Brooks et al., 2018*). In addition, a study conducted in Amsterdam showed that the relative abundance of *Christensenellaceae* is ethnicity specific (*Brooks et al., 2018*; *Deschasaux et al., 2018*). These studies demonstrated that age, gender, and ethnicity are associated with the abundance of *Christensenellaceae*. The relative abundance of *Christensenellaceae* in the intestine was found to be inversely proportional to body mass index (BMI) (*Fu et al., 2015*; *Goodrich et al., 2014*; *Oki et al., 2016*; *Peters et al., 2018b*) and exhibited a negative correlation with obesity and inflammatory bowel disease (*Braun et al., 2019*; *Gevers et al., 2014*; *Imhann et al., 2018*).

We previously performed the Guangdong Gut Microbiome Project (GGMP) study, constructing the largest intestinal microbiota database for Eastern countries to date. In the present study, we selected the *Christensenellaceae*-related population from GGMP to find out the relationship between *Christensenellaceae* and regional distribution, metabolic index and metabolic diseases. We also focused on the connection between sequential operational

taxonomic unit (sub-OTU) (*Amir et al., 2017*) members of *Christensenellaceae* and the metabolic index. We also utilized the GGMP dataset to reveal the metabolic pathway signatures of *Christensenellaceae*. A better understanding of the relationship between MetS and *Christensenellaceae* may lead to new therapeutic approaches for MetS.

## MATERIALS & METHODS

### Data collection and criteria for MetS

The data used in this analysis from the GGMP were described previously (*He et al., 2018b*). In brief, 9,172 individuals were investigated, and 6,896 people finished the survey and household sampling (2,276 missing metadata). By further filtering the metadata, some individuals with sequence number less than 10,000 were excluded from this analysis, so a total of 6,879 individuals were retained. The GGMP employs an in-person questionnaire to collect individual metadata. To evaluate how *Christensenellaceae* members affect the metabolism and gut microbes, we also compiled information on whether the individuals had diabetes, dyslipidemia, and pre-diabetic symptoms.

MetS was diagnosed as described by the Joint Committee for Developing Chinese Guidelines on Prevention and Treatment of Dyslipidemia in adults, based on three of the following five criteria for participants: (1) waist circumference >90 cm (male) or >85 cm (female), (2) fasting blood glucose (FBG) $\geq$ 6.1 mmol/L (110 mg/dl) or previously diagnosed with diabetes, (3) triglyceride (TG) $\geq$ 1.7 mmol/L (150 mg/dl), (4) high-density lipoprotein (HDL) <1.04 mmol/L (40 mg/dl), and (5) systolic/diastolic blood pressure (SBP/DBP) $\geq$ 130/85 mmHg or previously diagnosed with high blood pressure.

### Sample collection and DNA extraction

Stool samplers, ice bags and ice boxes were provided to collect and store samples after the questionnaire survey. After defecation, each participant recorded their Bristol stool score and stored the sample in an ice bag. All the samples were stored in a freezer ($-18\ °C$ to $-20\ °C$) for less than 3 days and then transported to the research laboratory (Guangdong CDC) in a cold-chain vehicle to maintain a low-temperature environment. Samples were transported and stored at the research laboratory in $-80\ °C$ freezers until further processing.

A total of 200 mg of each fecal sample was used for DNA extraction using the Fecal DNA Bead Isolation Kit (Bioeasy, Shenzhen) according to the manufacturer's instructions. Before the specimens were submitted to laboratory analysis, we prepared external standards to control for potential batch effects because multiple technicians and machines were involved in sample processing. Briefly, fecal samples were collected from three donors. For each donor, the samples were manually homogenized to obtain an even mixture, divided into 200 tubes and stored at $-80\ °C$. All stool samples were processed with identical protocols, including three external standards for each batch. For each DNA sample, the bacterial 16S rRNA gene was amplified with the following barcoded primers (shown from 5′ to 3′): V4F (GTGYCAGCMGCCGCGGTAA) and V4R (GGACTACNVGGGTWTCTAAT) (*Walters et al., 2016*). The primers contained Illumina adapters and a unique 8-nucleotide barcode. The PCR conditions included an initial denaturation at 94 °C for 5 min; 30 cycles of

denaturation at 94 °C for 30 s, annealing at 52 °C for 30 s, and elongation at 72 °C for 45 s; and a final extension at 72 °C for 5 min. The products were submitted for next-generation sequencing on an Illumina HiSeq 2500 platform using 500-cycle version 2 reagent kits (Beijing Genome Institute, BGI, Beijing).

## Microbiome bioinformatic analysis

Raw sequence data were managed and analyzed using Quantitative Insights Into Microbial Ecology software (QIIME, version 1.9.1) (*Caporaso et al., 2010*), and the sequences with Phred quality scores below 20 were then discarded. The method for processing of sequences was identical to that described in our previous reports (*He et al., 2018b*). PCoA based on unweighted UniFrac distances comparing bacterial community structure of samples between G1 and G2. Permutational multivariate analysis of variance (PERMANOVA) was carried out to measure effect sizes and significance differences in beta diversity. The threshold of statistical significance was set at $P < 0.05$. We carried out multivariate association analyses with linear modeling (MaAsLin) as described by *Morgan et al. (2012)* to examine the relationship between sequential taxonomic units and each of the MetS diagnostic factors and several related diseases. Age was used as a confounder, and the false discovery rate was limited to 0.05. The R package (ggtree) (*Yu et al., 2018*) was used to visualize the evolution tree data after multi-sequence comparison with QIIME. Data plotting and statistical analyses were performed by R (3.2.2) statistical software.

## Ethics approval and consent to participate

The present study was approved by the Ethical Review Committee of the Chinese Center for Disease Control and Prevention under approval notice no. 201519-A. Written consent was obtained from all participants.

## Statistical analysis

The significance of differences between two groups was resolved by the Wilcoxon rank-sum test. Spearman's rank correlation test was applied to analyze the correlation between two variables. The chi-square test was utilized to compare the ratios of two groups. $P$ values less than or equal to 0.05 were considered significant. The Benjamini and Hochberg method was used to modulate the $P$ value for multiple hypotheses.

# RESULTS

## The overall gut *Christensenellaceae* configuration of people in the GGMP

Exploration of the effects of the gut microbiota requires data from studies performed with a regionalized study design, comprehensive sampling and standardized experimental protocols. The population included in the GGMP has been previously described (*He et al., 2018b*). In the GGMP project, 6896 individuals were included according to our entry criteria, which followed the guidelines of the Joint Committee for Developing Chinese Guidelines as mentioned in a previous article (*He et al., 2018*). In the GGMP study, totally 6896 samples were characterized by 16S rRNA gene sequencing, and more than 17,083 quality-filtered sequences were obtained through QIIME analyses. Based on our
statistics, the family *Christensenellaceae* accounted for an average of 0.11% of human fecal bacteria in the residents. Then, we grouped the individuals based on the abundance of *Christensenellaceae*; in total, 3316 people had no *Christensenellaceae* in their intestines (Group1, G1), and we selected the upper quartile population as Group2 (G2, $n = 1465$) according to *Christensenellaceae* abundance. The abundance of *Christensenellaceae* in this study was bias distributed, the overall median value was 0, and the maximum value was 15.1%. Statistics found that the average abundance of *Christensenellaceae* in the G2 group was 0.49%.To characterize the diversity and richness of the bacterial community, the alpha indices, estimated by four different parameters, were analyzed for each sample. As shown in Fig. S1, the diversity and richness estimators in the two cohorts were significantly different. Compared with the G1 subjects, the G2 subjects had a significantly high alpha diversity indexes, such as the Chao1, observed OTU, PD_whole tree and Shannon indexes ($p < 0.001$ for each). To measure the degree of similarity of the fecal microbial communities, we performed a principal coordinate analysis (Fig. 1A), and we found obvious differences between the two groups ($R2 = 0.07184, P = 0.001$). Additionally, we applied linear discriminant analysis effect size (LEfSe) (*Segata et al., 2011*) for quantitative analysis of biomarkers within different cohorts. A total of 20 features had significantly different abundances between G1 and G2(LDA>3) (Fig. 1B). At the genus level, the fecal microbiota of people who lacked *Christensenellaceae* was enriched with the taxa *Proteobacteria, Enterobacteriales, Bacteroidaceae, Lachnospiraceae* and *Klebsiella*, whereas individuals with a high proportion of *Christensenellaceae* exhibited enrichment of *Ruminococcaceae, Mollicutes, RF39, Akkermansia* and *Rikenellaceae*.

To examine the *Christensenellaceae* components in detail, we analyzed the sub-OTUs at the family level. After chimera removal and quality filtering, a total of 134 different sub-OTUs of *Christensenellaceae* were identified at the 100% sequence identity level, and eight of them had read counts greater than 1,000 (Fig. 1C). An evolutionary tree was constructed to further observe the evolutionary distance of each sub-OTU and several reference genome (Fig. S1).

## The distribution of *Christensenellaceae* in GGMP

In our survey for the GGMP study, the distribution of *Christensenellaceae* in Guangdong Province varied based on geographic location. According to the average abundance data, *Christensenellaceae* is unevenly geographically distributed, with the abundance typically reduced in or near large and economically thriving centers such as Guangzhou (0.08%) and Shenzhen (0.09%). In contrast, Shanwei had the highest abundance (0.22%), followed by Huizhou (0.19%) and Meizhou (0.18%). The abundances in other cities is shown in Fig. 2A. In addition, we collected data on per capita annual income and *Christensnellaceae* abundance in each region (Fig. 2B). According to the map, the higher the level of urbanization, the lower the abundance of *Christensenellaceae* in the intestinal tract of urban residents.

When the MetS prevalence in different cities was compared, there was a significantly negative correlation between MetS and *Christensenellaceae* abundance (Fig. 2C). Residents in Zhanjiang, Shanwei, Foshan and Maoming had *Christensenellaceae* abundances higher

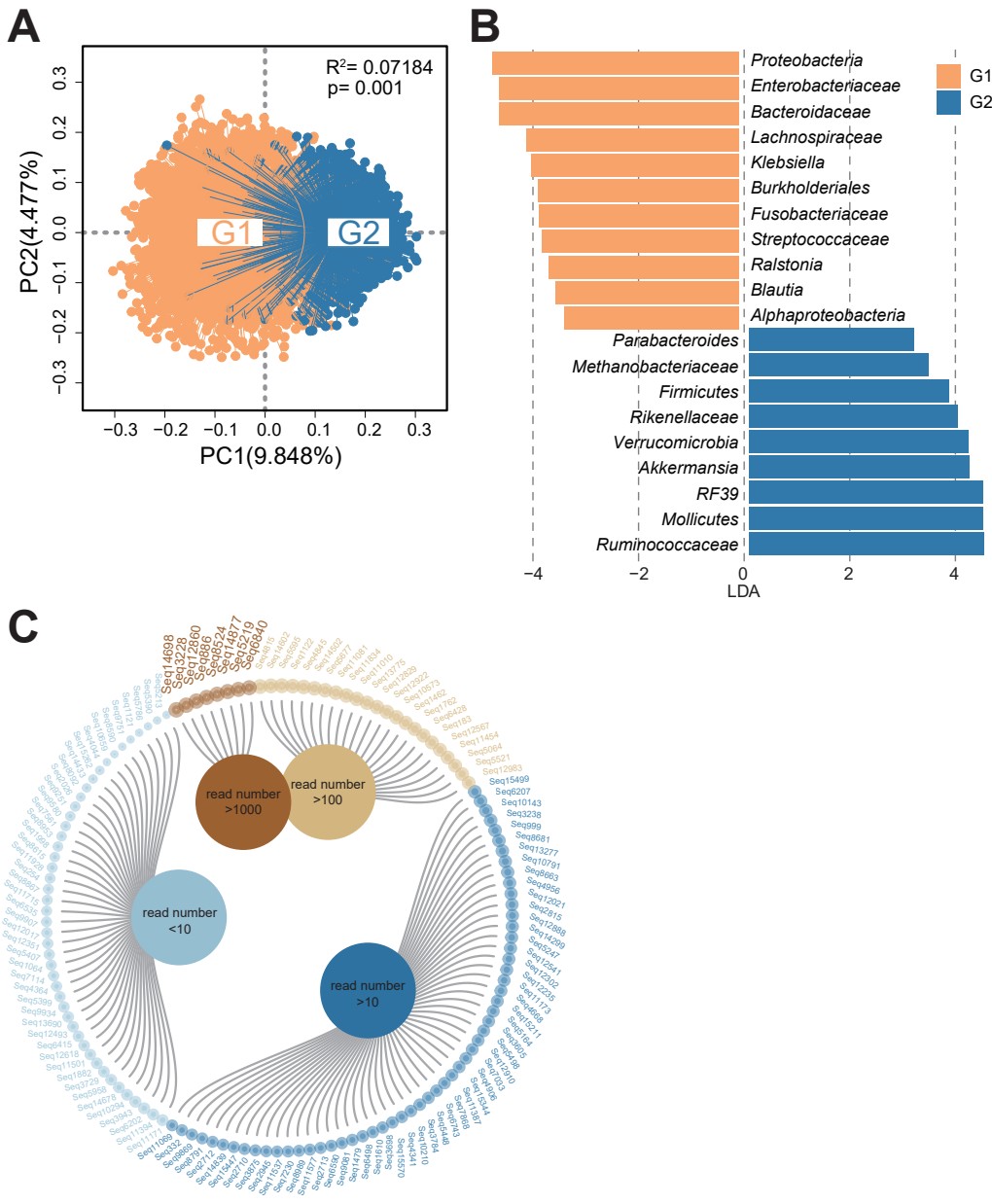

**Figure 1** Bacterial community structure analysis and sequential operational taxonomic units of *Christensenellaceae.* (A) PCoA based on unweighted UniFrac distances comparing bacterial community structure of samples between G1 and G2. Permutational multivariate analysis of variance (PERMANOVA) was carried out to measure effect sizes and significance differences in beta diversity. The threshold of statistical significance was set at $P < 0.05$. (B) Visualization of taxa meeting a LDA threshold > 3. LEfSe cladogram showed the differential abundant taxa between the two cohorts. Subjects without *Christensenellaceae* in orange; *chris*-rich subjects in blue. (C) Sub-OTUs at the level of *Christensenellaceae* were arranged according to the read number.

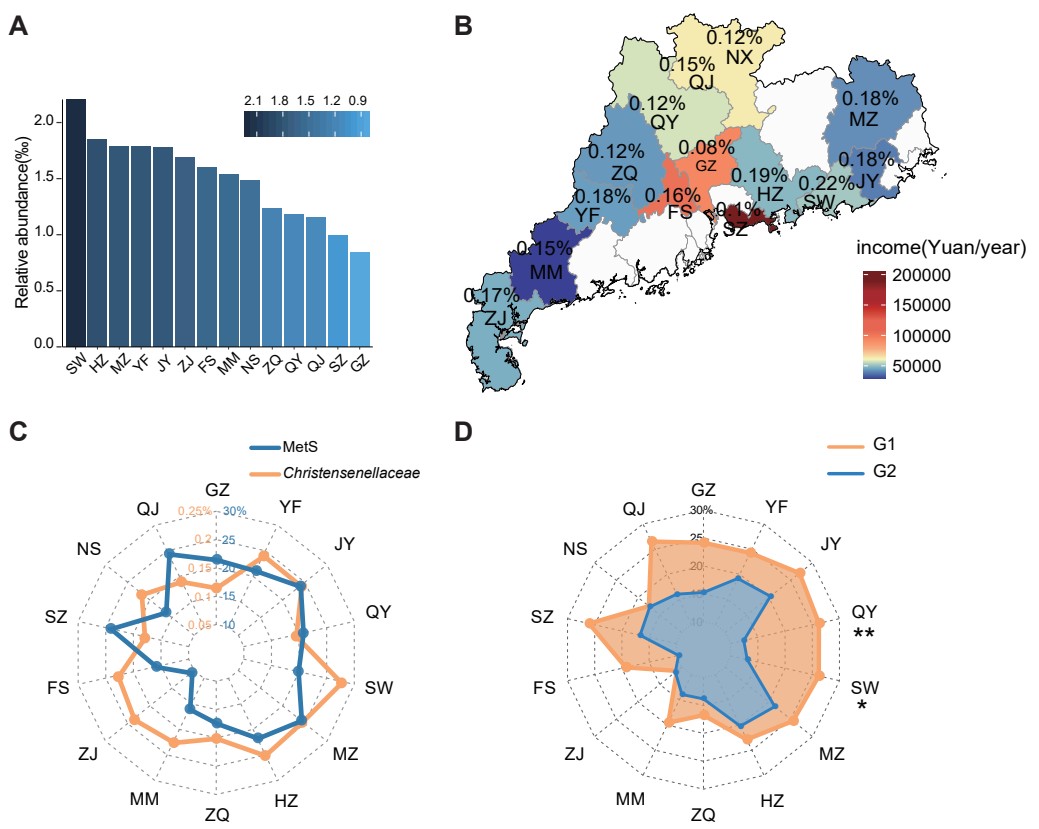

**Figure 2  The geographical features of *Christensenellaceae* in different regions in GGMP.**  (A) Histogram based on the average abundance of *Christensenellaceae* in 14 regions of GGMP. City abbreviation is on the *X*-axis and the abundance value in on the *Y*-axis. (B) Annual income and *Christensenellaceae* abundance in 14 regions of GGMP. (C) The prevalence of metabolic syndrome and the average abundance of *Christensenellaceae* in each region. The blue line represents metabolic syndrome prevalence and the orange line represents average abundance of *Christensenellaceae*. (D) Prevalence of metabolic syndrome in people with different *Christensenellaceae* abundance in each area. Light orange represents G1 ($n = 3316$) and light blue represents G2 ($n = 1465$).

than 0.15% in their gut and exhibited a low prevalence of MetS. Similarly, city dwellers in Guangzhou, Shenzhen and Shaoguan had a low *Christensenellaceae* abundance, and a high proportion of the population exhibited MetS. However, a negative correlation was not observed in residents in Jieyang and Meizhou. Notably, the prevalence of MetS in G2 subjects was lower than that in G1subjects in most regions (Fig. 2D). Moreover, there was a significant difference in the prevalence of MetS in people with different abundances of *Christensenellaceae* in Qingyuan ($p < 0.01$) and Shanwei ($p < 0.05$). Although *Christensenellaceae* abundance varies in different regions, the negative correlation between MetS and *Christensenellaceae* is universal.

## Christensenellaceae is associated with host metabolic index and MetS status

To explore the relationship between *Christensenellaceae* abundance and human health, we next assessed how variable the body parameters were in terms of *Christensenellaceae* richness. We selected 4,781 subjects associated with *Christensenellaceae* from the GGMP project, and subjects in this study cohort had a wide range of age, BMI and blood test indicators (Table 1). As previously described, we classified the subjects into two categories (G1 and G2) according to the abundance of *Christensenellaceae*. The differences between the two groups were mainly reflected in the following aspects: age, anthropometric parameters and biochemical criteria (Table 1). Because variations in metabolites could be related to differences in age (*Dunn et al., 2015*), we reanalyzed the correlation between the indices after adjusting for age. Compared to the G1 group, people rich in *Christensenellaceae* showed significantly low BMI, waist circumference and waist-to-height ratio (WHtR). The high-value group also showed lower levels of biochemical indices, such as TG, alanine transaminase (ALT) and uric acid (UA), than the non-*Christensenellaceae* group (Fig. S2). The level of HDL was significantly higher in people with *Christensenellaceae*, indicating the abundance of *Christensenellaceae* has a positive correlation with HDL level.

In addition, we calculated the correlation coefficients between the waist circumference, WHtR, BMI, TG level and other clinical parameters of all subjects to evaluate the connection intensity between the abundance of *Christensenellaceae* and the host metabolic index (Fig. 3A). We found that waist circumference and WHtR remained significantly decreased in people rich in *Christensenellaceae* compared to the people who lacked *Christensenellaceae*. Recently, waist circumference and WHtR were proposed as predictors of the incidence of MetS (*Perona et al., 2019*; *Suliga et al., 2019*). Therefore, we measured the prevalence of MetS in individuals with GGMP. People rich in *Christensenellaceae* had low prevalence of metabolic diseases, including overweight, obesity, fatty liver disease, and hypertriglyceridemia.

We also tested for connections between the sequential taxonomic units of *Christensenellaceae* and the indicators to observe whether the changes in sub-OTUs at the family level of *Christensenellaceae* are consistent. These top eight sub-OTUs, with reads number greater than 1,000, had the same correlation with each parameter (Fig. 3A). They all had negative correlations with ALT, TG, UA, waist circumference and WHtR. Among the sub-OTUs, sub-OTU3228 had a significantly negative correlation with TG, ALT, UA and BMI. Nevertheless, most of the taxonomic units were positively related to HDL and blood urea nitrogen (BUN).

In addition to morphological indicators and circulating metabolites, we also identified an association between the family *Christensenellaceae* and metabolic diseases such as fatty liver disease, obesity, hypertriglyceridemia and digestive system disease (Fig. 3B). Moreover, we found that sub-OTU3228, sub-OTU5291, sub-OTU12860, and sub-OTU14698 were negatively correlated with overweight, obesity and hypertriglyceridemia. Although most of the associated taxa were shared across obesity and lipid metabolites, several sequences were predominantly linked to digestive diseases rather than metabolic disorders. Notably, the abundances of sub-OTU3228 and sub-OTU14698 were negatively associated with

**Table 1** Anthropometric parameters and biochemical criterions variations stratified by abundance of *Christensenellaceae.*

|  | *G1*-value (*n* = 3316) | *G2*-value (*n* = 1465) | *P*-value |
|---|---|---|---|
| Age(years) | | | |
| Median[SD] | 53.0[14.5] | 55.0[15.2] | <0.001 |
| Sex | | | |
| Female | 1830(55.2%) | 824(56.2%) | 0.517 |
| Male | 1486(44.8%) | 641(43.8%) | |
| BMI(kg/m$^2$) | | | |
| Median[SD] | 23.4[3.61] | 22.5[3.23] | <0.001 |
| Missing | 39(1.2%) | 27(1.8%) | |
| Waist(cm) | | | |
| Median[SD] | 80.4[10.1] | 78.3[9.22] | <0.001 |
| Missing | 39(1.2%) | 27(1.8%) | |
| WHtR | | | |
| Median[SD] | 0.512[0.0651] | 0.497[0.0603] | <0.001 |
| Missing | 39(1.2%) | 27(1.8%) | |
| Obesity | | | |
| Yes | 394(11.9%) | 92(6.3%) | <0.001 |
| No | 2883(86.9%) | 1346(91.9%) | |
| Missing | 39(1.2%) | 27(1.8%) | |
| MetS | | | |
| Yes | 739(22.3%) | 227(15.5%) | <0.001 |
| No | 2529(86.4%) | 1209(82.5%) | |
| Missing | 38(1.1%) | 29(2.0%) | |
| Hypertriglyceridemia | | | |
| Yes | 414(12.5%) | 99(6.8%) | <0.001 |
| No | 2864(86.4%) | 1339(91.4%) | |
| Missing | 38(1.1%) | 27(1.8%) | |
| UA(μmol/L) | | | |
| Median[SD] | 330[94.5] | 315[87.3] | <0.001 |
| Missing | 38(1.1%) | 27(1.8%) | |
| TG(mmol/L) | | | |
| Median[SD] | 1.15[1.56] | 0.9555[1.36] | <0.001 |
| Missing | 38(1.1%) | 27(1.8%) | |
| ALT(U/L) | | | |
| Median[SD] | 16.0[16.5] | 14.0[13.7] | <0.001 |
| Missing | 107(3.2%) | 52(3.5%) | |
| Hb(g/L) | | | |
| Median[SD] | 143[21.9] | 141[20.3] | <0.001 |
| Missing | 38(1.1%) | 27(1.8%) | |
| HDL(mmol/L) | | | |
| Median[SD] | 1.22[0.520] | 1.25[0.435] | 0.0155 |
| Missing | 40(1.2%) | 27(1.8%) | |

**Notes.**

SD, standard deviation; BMI, indicates body mass index; WHtR, waist-to-height ratio; UA, uric acid; TG, triglyceride; ALT, alanine transaminase; Hb, hemoglobin; HDL, high density lipoprotein.
digestive disorders, while sub-OTU3228 showed a strong positive correlation with the occurrence of diabetes mellitus (T2DM). Besides the above four members, the remaining taxonomic units with read counts greater than 1,000 were not significantly associated with these metabolism-related disorders.

## Specific microbial taxa associated with *Christensenellaceae*

We used co-occurrence network analysis to investigate the interactions among the microbes in the complex intestinal microbiota. After pairwise correlation analysis of the bacteria in all the volunteers' stool samples, the major bacteria associated with *Christensenellaceae* in the individuals enrolled in the study are shown in Fig. 4A. Of all the species, the abundances of *Veillonella, Ruminococcus, Fusobacterium,* and *Blautia* were most highly negatively related to *Christensenellaceae*. Additionally, species such as *Ralstonia* and *Klebsiella* had a negative correlation with the abundance of *Christensenellaceae*. In addition, we identified a significant positive association of *Christensenellaceae* with *Oscillospira, Ruminococcaceae, RF39, Rikenellaceae* and *Akkermansia*. Moreover, *Lachnospiraceae, Roseburia* and *Sediminibacterium* also showed expected relationship with *Christensenellaceae*.

The focus was not only on the family level of the microbiota but also on the sub-OTUs of *Christensenellaceae*. The correlation between sequential taxonomic units of *Christensenellaceae* and other bacteria was universal. As shown in Fig. 4B, the eight sequences with read counts in excess of a thousand were consistent with the relationships between other bacteria. The four sequences sub-OTU14698, sub-OTU3228, sub-OTU12860, and sub-OTU5219 showed the strongest correlation, and the predominant bacteria in the positive correlation included *Oscillospira, Clostridiales, RF39* and *Rikenellaceae*. Nevertheless, *Veillonella, Fusobacterium, Blautia, Megamonas* and *Streptococcus* were negatively associated with these five sub-OTUs. In general, the bacterial network relationship at the sequence level and the family level is basically the same.

## Functional properties predicted by PICRUSt

We performed PICRUSt analysis to predict the KEGG functional orthologs of the fecal microbiota metagenomes based on 16S rRNA sequences. Principal component analysis (PCA) showed that the KO profile of the gut microbiota in people with high abundance of *Christensenellaceae* diverged from that of the gut microbiota of people lacking *Christensenellaceae* (Fig. S3). 71.2% of the variation within these two groups was captured by the first principal component (PC1). As shown in Fig. 5, broad potential communication pathways were identified between the individuals, including metabolism, cellular process, environmental information process, genetic information process and human disease. The LEfSe algorithm was applied to detect differences in the functional pathways of the microbiota between the two groups. In total, 6 functional orthologs were significantly different in *Christensenellaceae*-rich people (LDA score>3); the enriched orthologs were nucleotide metabolism; cellular process; ribosome; translation; replication and repair; and genetic information processing. In contrast, the prevalent markers among the controls included those associated with transporters, membrane transport and environmental
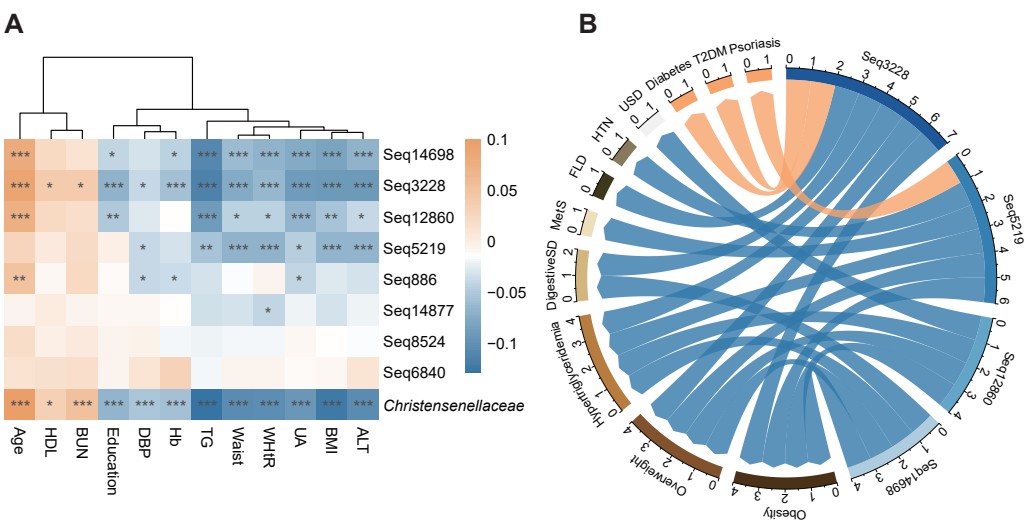

**Figure 3** **Metabolism related index and disease status related to *Christensenellaceae*.** (A) Heatmap of the association between metabolic index and *Christensenellaceae* as well as its sOTU. Color depth indicates the pearson correlation coefficients. \*\*\*$P < 0.001$, \*\*$P < 0.01$, \*$P < 0.05$. (B) The network showed the significant association between *Christensenellaceae* sub-OTUs and metabolic diseases. The blue arrow represents the negative correlation, and the orange represents positive. The number of significant association was summarized on the edge of network. Mets, metabolic syndrome; FLD, fatty liver disease; HTN, hypertension; USD, urinary system diseases; T2DM, type 2 diabetes.

information processing (Fig. 5). Moreover, several metabolic pathways were abundant in the non-*Christensenellaceae* group, including those involved in energy metabolism (methane metabolism), lipid metabolism (fatty acid biosynthesis) and carbohydrate metabolism (fructose and mannose metabolism) (Table S1). In terms of metabolic disease, people with *Christensenellaceae* have a relatively low risk of diabetes mellitus (Table S1). These findings suggest that *Christensenellaceae* may affect the way in which we metabolize and regulate ponderal growth.

# DISCUSSION

This study was based on a rigorous experimental design and strict quality control, and fecal samples and host information of 6,896 volunteers were collected in Guangdong Province. The use of 16S rRNA sequencing helped us clearly elucidate the complexity of the gut microbial ecosystem. We could focus not only on the composition and distribution of intestinal bacteria but also on the metabolic functions of these bacteria. Through the integration of these data, we could focus on the associated OTUs classified within the *Christensenellaceae* family. And we can target these OTUs as a keystone and link the etiology and pathogenesis of MetS to intestinal microorganisms to provide potential therapeutic targets.

First, the present study showed a variation in *Christensenellaceae* among residents in different areas by comparing the average abundance for each region. Based on our previous research, MetS prevalence was significantly higher in individuals with higher

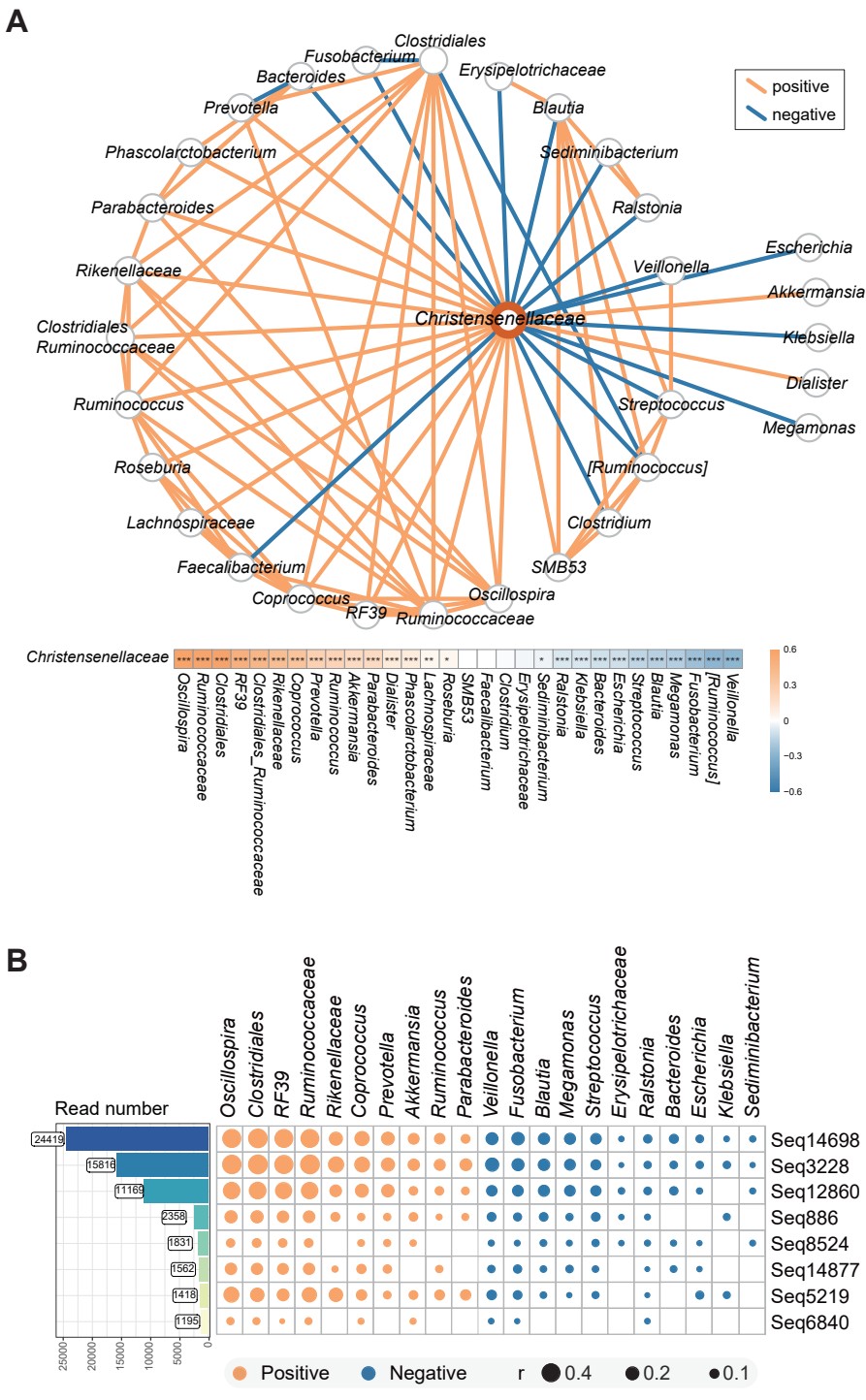

**Figure 4 Microbial taxa associated with *Christensenellaceae*.** (A) Co-occurrence network between *Christensenellaceae* and the relative microbiota. The orange line indicates positive correlation and blue negative correlation. ***$P < 0.001$, **$P < 0.01$, *$P < 0.05$. (B) Matrix Diagram of sub-OTUs within *Christensenellaceae* and related microbiota. Microbiota that are significantly associated with *Christensenellaceae* were plotted. The size of the plots represents the strength of the correlation.

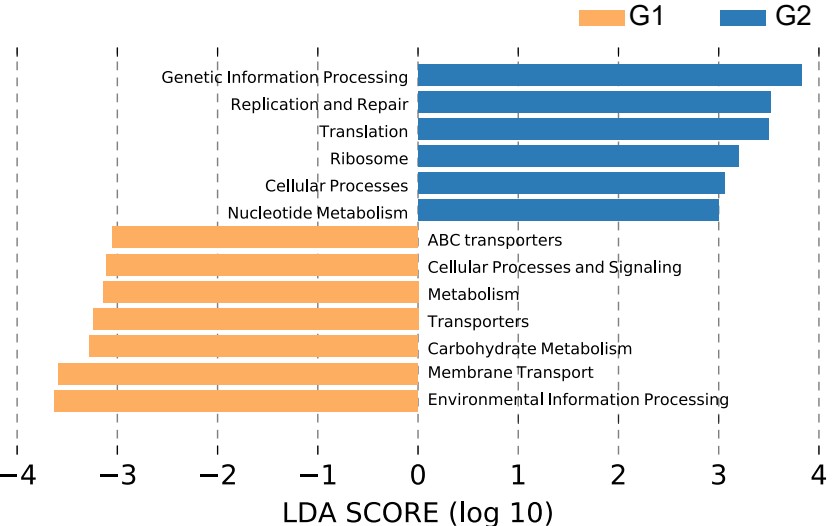

**Figure 5** **Differential pathways in individuals with different abundance of *Christensenellaceae*.** G1, LDA score < −3, orange color; G2, LDA > 3, blue color.

economic status than in those of with lower economic status (*He et al., 2018a*). As we found that *Christensenellaceae* is negatively correlated with MetS, it can be explained to some extent that people in economically developed regions have lower intestinal abundance of *Christensenellaceae*. Moreover, the individual differences in *Christensenellaceae* may simply be a result of environmental or genetic factors (*Waters & Ley, 2019*). And our results for large study population showed that the negative correlation between MetS and *Christensenellaceae* is consistent in most areas.

As previously reported, the abundance of *Christensenella, Bifidobacterium* and *Akkermansia* has been recognized as a signature of the gut ecosystem for healthy aging and longevity (*Biagi et al., 2016*; *Derrien, Belzer & De Vos, 2017*; *Wang et al., 2015*). We also found that people with a high abundance of *Christensenellaceae* were older than those in the control group. Changing functions of the intestine with age may affect the abundance of bacteria.

The microbiota inhabiting the intestinal cavity affects body health by altering the metabolome and regulating the bacterial bioavailability of nutrients in the lumen. (*Johnson et al., 2015*; *Liu et al., 2018*; *Romano et al., 2015*). In this study, we observed a significant association between the gut microbiota and the variation in BMI and blood lipid levels, which is independent of age. We observed that the abundance of *Christensenellaceae* was significantly related to the individual variance in BMI and to the blood levels of TG and HDL but has not much correlation with low-density lipoprotein (LDL) or total cholesterol (TC) levels. The analysis showed that people with higher *Christensenellaceae* abundance in their gut had lower BMI and lower TG levels. These results are consistent with the previous report demonstrating that *Christensenellaceae* abundance was negatively correlated with BMI and triglycerides and positively correlated with HDL levels in the Dutch LifeLines DEEP cohort ($n = 893$) and reports from other countries (*Fu et al., 2015*; *Peters et al., 2018a*;

*Waters & Ley, 2019*). As low HDL levels are one of the criteria of metabolic syndrome (*Yang & Wang, 2019*). The positive relationship between *Christensenellaceae* and HDL suggests a potentially beneficial role in metabolism. Therefore, in the complex network of indicators associated with *Christensenellaceae*, these findings show the negative correlation between *Christensenellaceae* with the prevalence of MetS.

The microbial interaction network has been considered an important biological factor in the occurrence and progression of metabolic system diseases. We observed significant microbial community changes in people with high abundance of *Christensenellaceae*, in whom the richness and diversity of the microbial community increased significantly. Our results highlight the positive correlation between *Christensenellaceae* and *Oscillospira, Ruminococcaceae, RF39, Rikenellaceae* and *Akkermansia* (*Goodrich et al., 2014*), while the bacteria were negatively related to *Veillonella, Fusobacterium* and *Klebsiella*. Previous studies have shown that *Oscillospira* and *Ruminococcaceae* are positively associated with health and leanness (*Konikoff & Gophna, 2016*; *Zietak et al., 2016*). Other noteworthy taxa include *Akkermansia, RF39* and *Rikenellaceae,* which have been experimentally confirmed to be probiotics and are considered beneficial to limit high fat-induced body weight gain. (*Alard et al., 2016*; *Wang et al., 2017*). Moreover, previous reports have shown that *Fusobacterium* and *Klebsiella* are positively correlated with the levels of cholesterol and LDL and alter lipid metabolism (*Fei et al., 2020*; *Koren et al., 2011*). How *Christensenellaceae* members impact the diversity and structure of the intestinal microbiota is still unclear. Nevertheless, it is plausible that changes in the intestinal niche induced by *Christensenellaceae* can promote lipid metabolism and contribute to the maintenance of normal body weight.

Our work showed differences in the predicted microbiota function in people with different abundances of *Christensenellaceae*. Previous reports indicated that obesity markers were typically positively associated with the KEGG categories of fructose metabolism (*Hannou et al., 2018*) and methane metabolism (*Mathur et al., 2013*), which are enriched in people who lack *Christensenellaceae*. However, it's not a general consistency with the previous result that *Methanobacteriaceae* increased in G2 group. PICRUSt represents the methane metabolism of the whole gut flora, but the single increased abundance of *Methanobacteriaceae* does not indicate the overall function. The high abundance of *Christensenellaceae* in individuals showed decreased fatty acid biosynthesis. Increasing evidence has shown that fatty acid accumulation is significantly associated with metabolic diseases and obesity (*Sonnenburg & Bäckhed, 2016*). This confirmed the previously observed negative correlation between *Christensenellaceae* and metabolic indicators. To develop a deeper understanding of MetS progression, further follow-up studies are required to examine the significance of microbial functional variations in body metabolism. Furthermore, metatranscriptomic and metabolomic analyses could be used to elucidate the detailed pathways of gene and metabolite interactions, enhancing the understanding of the effects of intestinal bacteria.

## CONCLUSIONS

In the present study, we used multivariate association analyses to explore the association between *Christensenellaceae* and metabolic indexes in nearly 5,000 participants in South

China. Based on the results, *Christensenellaceae* is related to a low risk of MetS and obesity, showing similarities to results obtained in Western populations. In addition, significant correlations were found between *Christensenellaceae* and other intestinal bacteria through analysis of the bacterial network. The bacteria that were positively correlated with *Christensenellaceae* were mainly beneficial bacteria that had been identified previously, while some pathogenic bacteria were negatively correlated with *Christensenellaceae*. Finally, PICRUSt was applied to analyze the pathway differences caused by *Christensenellaceae*, which provided bioinformatics evidence for predicting the correlation between *Christensenellaceae* and metabolic alterations in the gut microbiota community. However, the underlying mechanisms through which *Christensenellaceae* members regulate host metabolism need to be explored.

## ACKNOWLEDGEMENTS

We acknowledge the contributions of the 308 local CDC investigators and registered nurses, who helped with collection point maintenance and metadata and stool sample collection. We thank all the volunteers who participated in this project.

### Funding
This work was supported by the National Natural Science Foundation of China (No. 81925026). The funders had no role in study design, data collection and analysis, decision to publish, or preparation of the manuscript.

### Grant Disclosures
The following grant information was disclosed by the authors:
National Natural Science Foundation of China: 81925026.

### Competing Interests
The authors declare there are no competing interests.

### Author Contributions
- Xiang Li conceived and designed the experiments, performed the experiments, analyzed the data, prepared figures and/or tables, authored or reviewed drafts of the paper, and approved the final draft.
- Zewen Li conceived and designed the experiments, performed the experiments, analyzed the data, prepared figures and/or tables, and approved the final draft.
- Yan He analyzed the data, authored or reviewed drafts of the paper, and approved the final draft.
- Pan Li analyzed the data, prepared figures and/or tables, and approved the final draft.
- Hongwei Zhou and Nianyi Zeng conceived and designed the experiments, authored or reviewed drafts of the paper, and approved the final draft.

## Human Ethics

The following information was supplied relating to ethical approvals (i.e., approving body and any reference numbers):

The present study was approved by the Ethical Review Committee of the Chinese Center for Disease Control and Prevention under approval notice No. 201519-A. Written consent was obtained from all participants.

## Data Availability

The collated data of this analysis are available as Table S2.

The raw sequence data for the 16S rRNA gene are available from the European Nucleotide Archive: PRJEB18535.

The bioinformatics code used for raw data processing is available at Github: https://github.com/SMUJYYXB/GGMP.

## Supplemental Information

Supplemental information for this article can be found online at http://dx.doi.org/10.7717/peerj.9591#supplemental-information.

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
