# Peer review of "Regional distribution of Christensenellaceae and its associations with metabolic syndrome based on a population-level analysis"

_PeerJ, doi:10.7717/peerj.9591_

## Round 0.1 · original submission · Major Revisions

After an initial review of your manuscript, I found that there were quite a places where the wording prevented a complete understanding of the science. Therefore, in order to be able to send this manuscript out for review, we request that it first be professionally edited to substantially improve the grammar. Once it has been edited, we would be happy to send it out for review. Thank-you

---

## Round 0.2 · Minor Revisions

Overall, the reviewers of this manuscript found that the data and analyses presented will represent a useful contribution to the field, providing important information on the association of Christensenellaceae species with metabolic syndrome. They provide a number of suggestions for improving the manuscript that should be considered in a revised version. In particular, please address the concerns with regards to the statistical analyses and the methods used for, and inclusion of known Christensenellaceae species and strains in the phylogenetic tree. Finally, all of the raw data should be made available from the appropriate public repository such as the NCBI SRA, BioSample, and BioProject archive, with IDs linking to the repository records provided in the manuscript.

Reviewer 1 ·

Basic reporting

Comments:
Please review to improve clarity as noted below and see suggested figure modifications.
Line 17: change to ‘gut microbiota’
Line 23: “…analyzed by various bioinformatic methods” is ambiguous. Please restate or remove.
Line 25: “…a greater complexity of the co-occurrence network” is unclear. Please restate your meaning. More taxa co-occurred with Christensen ellaceae? Greater than what?
Line 32-33: “…which provides a good explanation for the negative relationship between body weight and Christensenellaceae.” Overstatement of correlative results; please reword to “which may explain the negative relationship.”
Line 37: “…application of Christensenellaceae as a probiotic.” Christensenellaceae is a broad and diverse family of bacteria, whereas probiotics are delivered as strains. Please restate as “investigation of strains within Christensenellaceae as potential probiotics” or similar.
Line 53: “Specific microbial contributions…” Microbial metabolites?
Line 60-61: where the type strain Christensenella minuta was first isolated from…
Line 80: The term sOTU commonly refers to ‘sub-OTU.’ Please provide a citation to the definition of ‘sequential’ OTU if that is your meaning, or modify to ‘sub-OTU’ throughout
Line 196: Please rephrase – suggests Christensenellaceae was a treatment

Figure 3B: Needs additional description to what this represents. Please indicate if only significant correlations are shown and what p-value. What do the numbers and colors on the circle mean?
Figure 4B: Would benefit from a legend that indicates the range of values the dots represent
Figure S1 E: Needs units – distance or % similarity? Why do four of the sOTUs appear to be 100% identical?

Experimental design

Comments:
Line 89: Clarify the criteria process for removal/inclusion. 6896 people were included (some removed for missing metadata?), then in Line 101 you state 6879 (additional missing metadata?)
Line 93: ‘diabetes and early symptoms’ (pre-diabetic?)
Line 112: Explain what is the ‘external standard’
Line 124: ‘quality scores were then discarded’ is unclear. If you mean filtered by the Phred quality score, then please state the threshold (e.g. q20).

Validity of the findings

Comments:
Line 152: Please check “17083 quality-filtered sequences were obtained.” Does that refer to the average sequences per individual? Or number of Christensenellaceae sequences? Could you add the average Christensenellaceae abundance of the Q4 group?
Line 163: Please state statistical test or R package in methods that generated the R2 and p-value (e.g. PERMANOVA)
Line 172: units = sOTUs? Please keep consistent throughout.
Line 174: Please describe the software or tool used to construct the evolutionary tree in the Methods section.
Line 211-212: Overstatement of correlative results. Please rephrase.
Line 216-218: More appropriately moved to the Discussion
Line 221: Add reference
Line 227: Make clear that these are the most abundant sOTUs, with reads greater than 1000
Line 293: Please reword. The methods applied are unlikely to resolve strains.
Line 305: The mean age between the groups was 2 years difference, and overall not an ageing or elderly population. This conclusion would be better supported if age were used in the correlation analyses (Fig 3A).
-Please check for inappropriate usage of term ‘probiotic.’ See Hill, C., Guarner, F., Reid, G. et al. The International Scientific Association for Probiotics and Prebiotics consensus statement on the scope and appropriate use of the term probiotic. Nat Rev Gastroenterol Hepatol 11, 506–514 (2014) for criteria.
-The lower abundance of Christensenellaceae found in urban populations is interesting, and not well-addressed in the discussion. Do the authors find any literature to support this finding? Are there obvious lifestyle or dietary factors that may help to explain?

Additional comments

This manuscript represents a sub-analysis of a previously published dataset from the GGMP study. The strength of this study is the very large subject number and sampling that occurred from different geographic regions in China. The analyses completed are straight-forward and highlight interesting correlations between the abundance of Christensenellaceae and MetS. A limitation of the study is that it is observational, and the associations to metabolic health are correlative. The authors have identified some potential targets that may warrant further testing in efficacy-type animal models.

Reviewer 2 ·

Basic reporting

This manuscript is clearly written although there are some spelling and grammar mistakes to correct. The literature referenced needs to be completed as listed below.

The raw data are not presently listed and should be added prior publication.


INTRODUCTION:

The introduction misses to reference a number of important publications.
Line 50: the reference 2019 is missing
Line 53: on the topic of microbiota, diet and gene interactions, the excellent review by Zmora et al. published in 2019 in Nat Rev Gastro Hepatol could be added along Ussar et al 2015
Line 62: please replace the reference to the review (Waters and Ley) by the original work by Morotomi and co-authors (2012) who named the new Christensenellaceae family after Prof Henrik Christensen and correct spelling (not Henkry)


Please replace the word “race” by “ethnicity” lines 70 and 71. The link with ethnicity is actually very well illustrated by the study by Brooks et al 2018. This reference should also be mentioned along Deschasaux et al.

Line 73: relationships with BMI have been reported by many other studies. Add at least a few such as Peters et al. 2018, Fu et al. 2015 Circulation Research, Oki et al. 2016 BMC Microbiology

Line 74: evidence of associations between Christensenellaceae and IBD have also been reported more recently by Braun et al. 2019 in the American Journal of Gastroenterology

FIGURE 1D: Cladogram is too small to be read; increase size or remove

TABLE 1: Units are missing

SUPPLEMENTARY FILES:

A list of concomitant medication in the metadata table provided would be very useful

Experimental design

Major comments:

1- To perform a comparative study of 2 groups, the authors chose to divide the cohort between a group where no Christensenellaceae were detectable, named Q1, and another extreme of the population distribution, named Q4. The authors describe these 2 groups as 2 quartiles (lower and upper quartiles of the distribution), although these actually do not correspond to quartiles. The population stratification must therefore be clarified. In addition, a better description of the Christensenellaceae distribution should be added, including overall median value, standard deviation, maximum value…

2- The authors identified several OTUs within the Christensenellaceae family that they used to make an evolutionary tree. Yet, the evolutionary tree does not include any reference V4 16S of the known Christensenellaceae members. In particular, the tree should include the reference genome of C. minuta strain DSM22607, which is the original type species strain of reference. Other species to include are C. timonensis, C. massiliensis and C. hongkongensis.

The Christensenellaceae OTUs sequences must be published along the paper since they form a major part of the present manuscript.

3- The authors report interesting variations between various geographical locations within Southern China that they relate to variations in “economically thriving centres”. A map identifying each region showing such variations in Christensenellaceae along a marker of economical health, such as the degree of urbanisation, would be very useful to appreciate this result.

Minor comments in MATERIAL AND METHODS:

Line 92: clarify what “metabolic index” means; should this be replaced by “metabolism”?
Line 93: diabetes is cited twice

Validity of the findings

A few sentences in the results and discussion are overstated or in contradiction with some results as listed below:

RESULTS:

Line 196: change title to avoid overstatement. Correlation is not causality.

Line 211: this is also overstated, especially since there is no indication of an impact on LDL or on HDL/LDL ratio.

Line 281: “methane metabolism”; this is in contradiction with fig 1 that shows methanogens to be increased in Q4. Can the authors comment on this in their discussion?

Line 283: “low risk of diabetes mellitus”; this is contradicted by figure 3B showing a positive correlation between some OTUs of Christensenellaceae and T2D



DISCUSSION:

Line 293: “strains” and “species”; no strains nor species were studied in this work; replace “strains” by “OTUs classified within the Christensenellaceae family”; “key species” could be replaced by “keystone taxon”

Line 295: “metatranscriptomic and metabolomic analyses…” ; Neither metatranscriptomic, nor metabolomic analyses were performed in this work, so I suppose the authors are making some suggestions for future work here. Therefore this statement should be moved at the very end of the manuscript as a suggestion for future work.

Line 311: “contributes”; no causation relationship was evidenced in this study; this should be replaced by something along the line of “was significantly associated with…”

Line 315: Reference to Fu et al. 2015 Circulation Research could be added here along Peters and Waters

Line 325: Oscillospira and Ruminococcaceae have been regularly reported as positively correlated with Christensenellaceae in other studies such as Goodrich et al 2014

Line 338-340: This is an overstatement; the PICRUST analysis only predicts biochemical pathways potentially modulated in the microbiota population, not in the host. Therefore this statement is a shortcut between microbial metabolism and potential host targeted effects. mTOR stands for “mammalian Target Of Rapamycin”… This is a perfect illustration of an abusive use of the KEGG database by the Picrust software. The authors must adjust their analysis, either to remove pathways that are not found in microbes, or by adding a statement in the discussion to explain this odd association and remove reference to endogenous metabolic regulation.

CONCLUSION:

Line 353: “probiotics” is not fully appropriate here since many of the bacteria it refers to have not been developed as food. Suggested change is to replace with “beneficial bacteria”

Additional comments

Minor comments:

Line 152: “17083 quality-filtered sequences”: is this per individual?
Line 167: replace “genera” by taxa
Line 254: what is a “normal” relationship? Replace normal by “expected”
Line 270: “The first principal component…” This statement is incorrect. This only means that 71.20% of the variation in the dataset is captured by PC1. “Between” can be replaced by “within” to make this correct.
Line 275: suggestion to replace “altered” by “different”
Line 308: what is “bioavailability” referring to? This should be clarified
Line 323: suggested change: “microbiota was” by “bacteria were”
Line 327: this sentence is unclear; suggested change: replace “for host fat consumption” with “to limit high fat-induced body weight gain”
Line 330: “block” could be replaced with “alter”
Line 356: suggestion: replace “indicators” by “alterations in the gut microbiota community”

---

## Round 0.3 · Minor Revisions

Your manuscript just needs one minor modification before we will be able to accept it for publication. As suggested by the reviewer, please consider further modifying the discussion on the association of Christensenellaceae and HDL levels. Thanks.

Reviewer 2 ·

Basic reporting

This is a re-review. I will only comment on specific points already raised in review 1.

All points raised here in review 1 have been addressed.

Experimental design

All points raised here in review 1 have been addressed.

Validity of the findings

Line 211: this is also overstated, especially since there is no indication of an impact on LDL or on HDL/LDL ratio.
Response: We have restated the result to “The level of HDL was significantly higher in people with Christensenellaceae, indicating that the bacteria was negatively correlated with blood lipid level.”

This response is not satisfactory since an association with increased HDL is not an indication of a negative correlation with circulating lipid levels. I suggest to rather focus on the fact that low HDL levels are one of the criteria of metabolic syndrome and therefore, a positive correlation with HDL is indicative of a healthier metabolism.

Additional comments

Thank you for your consideration of my comments and for adjusting the manuscript accordingly.

---

## Round 0.4 · accepted · Accept

Thank-you for the submission of your revised manuscript and for addressing all of the reviewer's concerns. I am happy to now accept your manuscript for publication.